# Learning to Scaffold:
# Optimizing Model Explanations for Teaching

**Patrick Fernandes**[*,Ψ,Ω,ℜ]     **Marcos Treviso**[*,Ω,ℜ]     **Danish Pruthi**[†,Λ]
**André F. T. Martins**[Ω,ℜ,Γ]     **Graham Neubig**[Ψ]

[Ψ]Language Technologies Institute, Carnegie Mellon University, Pittsburgh, PA
[Ω]Instituto Superior Técnico & LUMLIS (Lisbon ELLIS Unit), Lisbon, Portugal
[ℜ]Instituto de Telecomunicações, Lisbon, Portugal
[Λ]Amazon Web Services     [Γ]Unbabel, Lisbon, Portugal

## Abstract

Modern machine learning models are opaque, and as a result there is a burgeoning academic subfield on methods that *explain* these models' behavior. However, what is the precise goal of providing such explanations, and how can we demonstrate that explanations achieve this goal? Some research argues that explanations should help *teach* a student (either human or machine) to simulate the model being explained, and that the quality of explanations can be measured by the simulation accuracy of students on unexplained examples. In this work, leveraging meta-learning techniques, we extend this idea to *improve the quality of the explanations themselves*, specifically by optimizing explanations such that student models more effectively learn to simulate the original model. We train models on three natural language processing and computer vision tasks, and find that students trained with explanations extracted with our framework are able to simulate the teacher significantly more effectively than ones produced with previous methods. Through human annotations and a user study, we further find that these learned explanations more closely align with how humans would explain the required decisions in these tasks. Our code is available at https://github.com/coderpat/learning-scaffold.

## 1  Introduction

While deep learning's performance has led it to become the dominant paradigm in machine learning, its relative opaqueness has brought great interest in methods to improve *model interpretability*. Many recent works propose methods for extracting *explanations* from neural networks (§ 7), which vary from the highlighting of relevant input features [Simonyan et al., 2014, Arras et al., 2017, Ding et al., 2019] to more complex representations of the reasoning of the network [Mu and Andreas, 2020, Wu et al., 2021]. However, are these methods actually achieving their goal of making models more interpretable? Some concerning findings have cast doubt on this proposition; different explanations methods have been found to disagree on the same model/input [Neely et al., 2021, Bastings et al., 2021] and explanations do not necessarily help predict a model's output and/or its failures [Chandrasekaran et al., 2018].

In fact, the research community is still in the process of understanding *what* explanations are supposed to achieve, and *how* to assess success of an explanation method [Doshi-Velez and Kim, 2017, Miller, 2019]. Many early works on model interpretability designed their methods around a set of desiderata [Sundararajan et al., 2017, Lertvittayakumjorn and Toni, 2019] and relied on qualitative assessment of a handful of samples with respect to these desiderata; a process that is highly subjective and is hard to reproduce. In contrast, recent works have focused on more quantitative criteria: correlation between

---

[*]  Equal contribution. Coresp. to `pfernand@cs.cmu.edu` or `marcos.treviso@tecnico.ulisboa.pt`
[†]  Work done while at Carnegie Mellon University, prior to joining Amazon.

36th Conference on Neural Information Processing Systems (NeurIPS 2022).

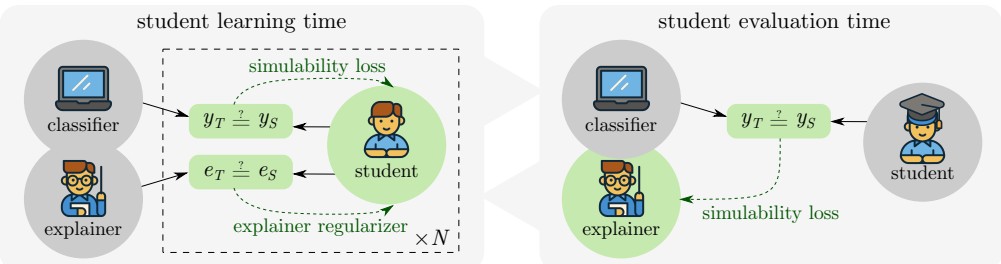

Figure 1: Illustration of our SMaT framework. First, a student model is trained to recover the classifier's predictions and to match the explanations given by the explainer. Then, the explainer is updated based on how well the trained student *simulates* the classifier (without access to explanations). In practice, we repeat these two consecutive processes for several steps. Green arrows and boxes represent learnable components.

explainability methods for measuring *consistency* [Jain and Wallace, 2019, Serrano and Smith, 2019], *sufficiency* and *comprehensiveness* [DeYoung et al., 2020], and *simulability*: whether a human or machine consumer of explanations understands the model behavior well enough to predict its output on unseen examples [Lipton, 2016, Doshi-Velez and Kim, 2017]. Simulability, in particular, has a number of desirable properties, such as being intuitively aligned with the goal of *communicating* the underlying model behavior to humans and being measurable in manual and automated experiments [Treviso and Martins, 2020, Hase and Bansal, 2020, Pruthi et al., 2020].

For instance, Pruthi et al. [2020] proposed a framework for automatic evaluation of simulability that, given a *teacher model* and explanations of this model's predictions, trains a *student model* to match the teacher's predictions. The explanations are then evaluated with respect to how well they help a student *learn to simulate* the teacher (§ 2). This is analogous to the concept in pedagogy of **instructional scaffolding** [Van de Pol et al., 2010], a process through which a teacher adds support for students to aid learning. More effective scaffolding—in our case, better explanations—is assumed to lead to better student learning. However, while this previous work provides an attractive way to *evaluate* existing explanation methods, it stops short of proposing a method to actually *improve* them.

In this work, we propose to *learn to explain* by directly learning explanations that provide better scaffolding of the student's learning, a framework we term *Scaffold-Maximizing Training* (**SMaT**). Figure 1 illustrates the framework: the explainer is used to *scaffold* the student training, and is updated based on how well the student does at *test* time at simulating the teacher model. We take insights from research on meta-learning [Finn et al., 2017, Raghu et al., 2021], formalizing our setting as a bi-level optimization problem and optimizing it based on higher-order differentiation (§ 3). Importantly, our high-level framework makes few assumptions about the model we are trying to explain, the structure of the explanations or the modalities considered. To test our framework, we then introduce a *parameterized* attention-based explainer optimizable with SMaT that works for any model with attention mechanisms (§ 4).

We experiment with SMaT in text classification, image classification, and (multilingual) text-based regression tasks using pretrained transformer models (§ 5). We find that our framework is able to effectively optimize explainers across all the considered tasks, where students trained with *learned* attention explanations achieve better simulability than baselines trained with *static* attention or gradient-based explanations. We further evaluate the *plausability* of our explanations (i.e., whether produced explanations align with how people would justify a similar choice) using human-labeled explanations (text classification and text regression) and through a human study (image classification) and find that explanations learned with SMaT are more plausible than the static explainers considered. Overall, the results reinforce the utility of scaffolding as a criterion for evaluating and improving model explanations.

## 2 Background

Consider a model $T : \mathcal{X} \to \mathcal{Y}$ trained on some dataset $\mathcal{D}_{\text{train}} = \{(x_i, y_i)\}_{i=1}^{N}$. For example, this could be a text or image classifier that was trained on a particular downstream task (with $\mathcal{D}_{\text{train}}$ being the training data for that task). *Post-hoc* interpretability methods typically introduce an *explainer* module $E_T : \mathcal{T} \times \mathcal{X} \to \mathcal{E}$ that takes a model and an input, and produces an explanation $e \in \mathcal{E}$ for the output of the model given that input, where $\mathcal{E}$ denotes the space of possible explanations. For instance,

interpretability methods using saliency maps define $\mathcal{E}$ as the space of *normalized* distributions of importance over $L$ input elements $e \in \triangle_{L-1}$ (where $\triangle_{L-1}$ is the $(L-1)$-probability simplex).

Pruthi et al. [2020] proposed an automatic framework for evaluating explainers that trains a *student* model $S_\theta : \mathcal{X} \to \mathcal{Y}$ with parameters $\theta$ to *simulate* the *teacher* (i.e., the original classifier) in a *constrained* setting. For example, the student can be constrained to have less capacity than the teacher by using a simpler model or trained with a subset of the dataset used for the teacher ($\hat{\mathcal{D}}_{\text{train}} \subsetneq \mathcal{D}_{\text{train}}$). In this framework, a baseline student $S_\theta$ is trained according to $\theta^* = \arg\min_\theta \mathbb{E}_{(x,y)\sim\hat{\mathcal{D}}_{\text{train}}}[\mathcal{L}_{\text{sim}}(S_\theta(x), T(x))]$, and its simulability $\text{SIM}(S_{\theta^*}, T)$ is measured on an unseen test set. The actual form of $\mathcal{L}_{\text{sim}}$ and $\text{SIM}(S_{\theta^*}, T)$ is task-specific. For example, in a classification task, we use cross-entropy as the simulation loss $\mathcal{L}_{\text{sim}}$ over the teacher's predictions, while the simulability of a model $S_{\theta^*}$ can be defined as the simulation accuracy, i.e., what percentage of the student and teacher predictions match over a *held-out* test set $\mathcal{D}_{\text{test}}$:

$$\text{SIM}(S_{\theta^*}, T) = \mathbb{E}_{(x,y)\sim\mathcal{D}_{\text{test}}}[\mathbb{1}\{S_{\theta^*}(x) = T(x)\}]. \tag{1}$$

Next, the training of the student is augmented with explanations produced by the explainer $E$. We introduce a student explainer $E_S : \mathcal{S} \times \mathcal{X} \to \mathcal{E}$, (the $S$-explainer) to extract explanations from the student, and *regularizing* these explanations on the explanations of teacher (the $T$-explainer), using a loss $\mathcal{L}_{\text{expl}}$ that takes explanations for both models:

$$\theta_E^* = \arg\min_\theta \mathbb{E}_{(x,y)\sim\hat{\mathcal{D}}_{\text{train}}} \left[ \underbrace{\mathcal{L}_{\text{sim}}\left(S_\theta(x), T(x)\right)}_{\text{simulability loss}} + \beta \underbrace{\mathcal{L}_{\text{expl}}\left(E_S(S_\theta, x), E_T(T, x)\right)}_{\text{explainer regularizer}} \right]. \tag{2}$$

For example, Pruthi et al. [2020] considered as a teacher explainer $E_T$ various methods such as LIME [Ribeiro et al., 2016], Integrated Gradients [Sundararajan et al., 2017], and attention mechanisms, and explored both attention regularization (using Kullback-Leibler divergence) and multi-task learning to regularize the student.

The key assumption surrounding this evaluation framework is that a student trained with *good* explanations should learn to simulate the teacher better than a student trained with bad or no explanations, that is, $\text{SIM}\left(S_{\theta_E^*}, T\right) > \text{SIM}\left(S_{\theta^*}, T\right)$. For clarity, we will refer to the simulability of a model $S_{\theta_E^*}$ trained using explanations as *scaffolded* simulability.

## 3 Optimizing Explainers for Teaching

As a **first contribution** of this work, we extend the previously described framework to make it possible to directly optimize the teacher explainer so that it can most effectively teach the student the original model's behavior. To this end, consider a *parameterized T-explainer* $E_{\phi_T}$ with parameters $\phi_T$, and equivalently a *parameterized S-explainer* $E_{\phi_S}$ with parameters $\phi_S$. We can write the loss function for the student and $S$-explainer as:

$$\mathcal{L}_{\text{student}}(S_\theta, E_{\phi_S}, T, E_{\phi_T}, x) = \mathcal{L}_{\text{sim}}\left(S_\theta(x), T(x)\right) + \beta\mathcal{L}_{\text{expl}}\left(E_{\phi_S}(S_\theta, x), E_{\phi_T}(T, x)\right). \tag{3}$$

While this framework is flexible enough to rigorously and automatically evaluate many types of explanations, calculating scaffolded simulability requires an optimization procedure to learn the student and $S$-explainer parameters $\theta, \phi_S$. This makes it non-trivial to achieve our goal of directly finding the teacher explainer parameters $\phi_T$ that optimize scaffolded simulability. To overcome this challenge, we draw inspiration from the extensive literature on meta-learning [Schmidhuber, 1987, Finn et al., 2017], and frame the optimization as the following bi-level optimization problem (see Grefenstette et al. [2019] for a primer):

$$\theta^*(\phi_T), \phi_S^*(\phi_T) = \arg\min_{\theta, \phi_S} \mathbb{E}_{(x,y)\sim\hat{\mathcal{D}}_{\text{train}}} \left[\mathcal{L}_{\text{student}}(S_\theta, E_{\phi_S}, T, E_{\phi_T}, x)\right] \tag{4}$$

$$\phi_T^* = \arg\min_{\phi_T} \mathbb{E}_{(x,y)\sim\mathcal{D}_{\text{test}}} \left[\mathcal{L}_{\text{sim}}\left(S_{\theta^*(\phi_T)}(x), T(x)\right)\right]. \tag{5}$$

Here, the *inner* optimization updates the student and the $S$-explainer parameters (Equation 4), and in the *outer* optimization we update the $T$-explainer parameters (Equation 5). **Importantly**, our framework does not modify the teacher, as our goal is to explain a model without changing its original behavior. Notice that we also simplify the problem by considering the more tractable simulation loss $\mathcal{L}_{\text{sim}}$ instead of the simulability metric $\text{SIM}(S_{\theta^*}, T)$ as part of the objective for the outer optimization.

Now, if we assume the explainers $E_{\phi_T}$ and $E_{\phi_S}$ are differentiable, we can use gradient-based optimization [Finn et al., 2017] to optimize both the student (with its explainer) and the $T$-explainer. In particular, we use *explicit* differentiation to solve this optimization problem. To compute gradients for $\phi_T$, we have to differentiate through a gradient operation, which requires Hessian-vector products, an operation supported by most modern deep learning frameworks [Bradbury et al., 2018, Grefenstette et al., 2019]. However, explicitly computing gradients for $\phi_T$ through a large number of inner optimization steps is computationally intractable. To circumvent this problem, typically the inner optimization is run for only a couple of steps or a *truncated* gradient is computed [Shaban et al., 2019]. In this work, we take the approach of taking a *single* inner optimization step and learning the student and $S$-explainer jointly with the $T$-explainer *without* resetting the student [Dery et al., 2021]. At each step, we update the student and $S$-explainer parameters as follows:

$$\theta^{t+1} = \theta^t - \eta_{\text{INN}} \nabla_\theta \, \mathbb{E}_{(x,y) \sim \hat{\mathcal{D}}_{\text{train}}} \left[ \mathcal{L}_{\text{student}}(S_{\theta^t}, E_{\phi_S^t}, T, E_{\phi_T^t}, x) \right] \tag{6}$$

$$\phi_S^{t+1} = \phi_S^t - \eta_{\text{INN}} \nabla_{\phi_S} \, \mathbb{E}_{(x,y) \sim \hat{\mathcal{D}}_{\text{train}}} \left[ \mathcal{L}_{\text{student}}(S_{\theta^t}, E_{\phi_S^t}, T, E_{\phi_T^t}, x) \right]. \tag{7}$$

After updating the student, we take an extra gradient step with the new parameters but only use these updates to calculate the *outer*-gradient for $\phi_T$, without actually updating $\theta$. This approach is similar to the *pilot update* proposed by Zhou et al. [2021], and we verified that it led to more stable optimization in practice:

$$\theta(\phi_T^t) = \theta^{t+1} - \eta_{\text{INN}} \nabla_\theta \, \mathbb{E}_{(x,y) \sim \hat{\mathcal{D}}_{\text{train}}} \left[ \mathcal{L}_{\text{student}}(S_{\theta^{t+1}}, E_{\phi_S^{t+1}}, T, E_{\phi_T^t}, x) \right] \tag{8}$$

$$\phi_T^{t+1} = \phi_T^t - \eta_{\text{OUT}} \nabla_{\phi_T} \, \mathbb{E}_{(x,y) \sim \mathcal{D}_{\text{test}}} \left[ \mathcal{L}_{\text{sim}} \left( S_{\theta(\phi_T^t)}(x), T(x) \right) \right]. \tag{9}$$

## 4   Parameterized Attention Explainer

As a **second contribution** of this work, we introduce a novel *parameterized* attention-based explainer that can be learned with our framework. Transformer models [Vaswani et al., 2017] are currently the most successful deep-learning architecture across a variety of tasks [Shoeybi et al., 2019, Wortsman et al., 2022]. Underpinning their success is the *multi-head attention mechanism*, which computes a *normalized* distribution over the $1 \leq i \leq L$ input elements in parallel for each head $h$:

$$A^h = \text{SOFTMAX}(Q^h(K^h)^\top), \tag{10}$$

where $Q^h = [q_0^h, \cdots, q_L^h]$ and $K^h = [k_0^h, \cdots, k_L^h]$ are the *query* and *key* linear projections over the input element representations for head $h$. Attention mechanisms have been used extensively for producing saliency maps [Wiegreffe and Pinter, 2019, Vashishth et al., 2019] and while some concerns have been raised regarding their faithfulness [Jain and Wallace, 2019], overall attention-based explainers have been found to lead to relatively good explanations in terms of *plausibility* and *simulability* [Treviso and Martins, 2020, Kobayashi et al., 2020, Pruthi et al., 2020].

However, to extract explanations from multi-head attention, we have two important design choices:

1. **Single distribution selection:** Since self-attention produces an attention matrix $A^h \in \triangle_{L-1}^L$, we need to *pool* these attention distributions to produce a single saliency map $e \in \triangle_{L-1}$. Typically, the distribution from a single token (such as [CLS]) or the *average* of the attention distributions from all tokens $1 \leq i \leq L$ are used.

2. **Head selection:** We also need to *pool* the distributions produced by each head. Typical ad-hoc strategies include using the mean over all heads for a certain layer [Fomicheva et al., 2021b] or selecting a single head based on plausibility on validation set [Treviso et al., 2021]. However, since transformers can have hundreds or even thousands of heads, these choices rely on human intuition or require large amounts of plausibility labels.

In this work, we approach the latter design choice in a more principled manner. Concretely, we associate each head with a weight and then perform a weighted sum over all heads. These weights are learned such that the resulting explanation maximizes simulability, as described in § 3. More formally, given a model $T_{\theta_T}$ and its query and key projections for an input $x$ for each layer and head $h \leq H$, we define a *parameterized, differentiable* attention explainer $E_{\phi_T}(T_{\theta_T}, x)$ as

$$s^h = \frac{1}{L} \sum_{i=1}^{L} (q_i^h)^\top K^h, \qquad E_{\phi_T}(T, x) = \text{SOFTMAX} \left( \sum_{h=1}^{H} \lambda_T^h s^h \right), \tag{11}$$

where the teacher's head coefficients $\lambda_T \in \triangle_{H-1}$ are $\lambda_T = \text{NORMALIZE}(\phi_T)$ with $\phi_T \in \mathbb{R}^H$.

In this formulation, $s^h \in \mathbb{R}^L$ represents the average *unnormalized attention logits* over all input elements, which are then combined according to $\lambda_T$ and normalized with SOFTMAX to produce a distribution in $\triangle_{L-1}$. We apply a normalization function NORMALIZE to head coefficients involved to create a *convex* combination over all heads in all layers. In this work we consider the sparse projection function NORMALIZE = SPARSEMAX [Martins and Astudillo, 2016], as:

$$\text{SPARSEMAX}(z) = \underset{p \in \triangle_{H-1}}{\arg\min} \|p - z\|_2.$$

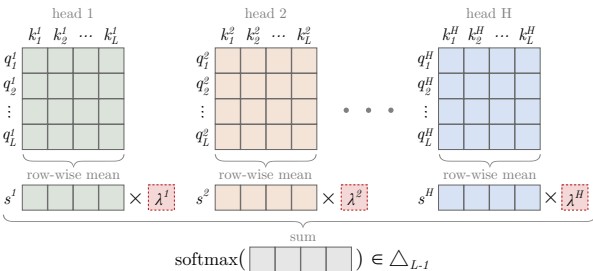

Figure 2: Our parameterized attention-based explainer. Dashed red boxes represent learned parameters $\lambda_T = \text{SPARSEMAX}(\phi_T) \in \triangle_{H-1}$, weighting average attention logits of each head $1 \leq h \leq H$. A softmax over the weighted sum generates the attention probabilities.

We choose SPARSEMAX due to its benefits in terms of interpretability, since it leads to many heads having zero weight. We also found it outperformed every other projection we tried (see § 6 for a more detailed discussion). Figure 2 illustrates each step of our parameterized attention explainer.

## 5 Experiments

To evaluate our framework, we attempt to learn explainers for transformer models trained on three different tasks: text classification (§ 5.1), image classification (§ 5.2), and machine translation quality estimation (a text-based regression task, detailed in § 5.3). We use JAX [Bradbury et al., 2018] to implement the higher-order differentiation, and use pretrained transformer models from the Huggingface Transformers library [Wolf et al., 2020], together with Flax [Heek et al., 2020]. For each task, we train a teacher model with AdamW [Loshchilov and Hutter, 2019] but, as explained in § 3, we use SGD for the student model (inner loop). We also use scalar mixing [Peters et al., 2018] to pool representations from different layers automatically.[3] We train students with a teacher explainer in three settings:

- **No Explainer**: No explanations are provided, and no explanation regularization is used for training the student (i.e. $\beta = 0$ in Equation 3). We refer to studentsin this setting as **baseline** students.
- **Static Explainer**: Explanations for the teacher model are extracted with five commonly-used saliency-based explainers: (1) L2 norm of gradients; (2) a *gradient × input* explainer [Denil et al., 2014]; (3) an *integrated gradients* explainer [Sundararajan et al., 2017]; and *attention* explainers that uses the *mean* pooling over attention from (4) all heads in the model and (5) from the heads of the last layer [Fomicheva et al., 2021b, Vafa et al., 2021]. More details can be found in Appendix A.
- **Learned Explainer (SMaT)**: Explanations are extracted with the explainer described in § 4, with coefficients for each head that are trained with **SMaT** jointly with the student. We initialize the coefficients such that the model is initialized to be the same as the *static* attention explainer (i.e. performing the mean over all heads).

Independently of the $T$-explainer, we always use a learned attention-based explainer as the $S$-explainer, considering all heads except when the $T$-explainer is a static attention explainer that only considers the last layers' heads, where we do the same for the $S$-explainer. We use the Kullback-Leibler divergence as $\mathcal{L}_{\text{expl}}$, and we set $\beta = 5$ for attention-based explainers and $\beta = 0.2$ for gradient-based explainers (since we found smaller values to be better). We set $\mathcal{L}_{\text{sim}}$ as the cross-entropy loss for classification tasks, and as the mean squared error loss for text regression. For each setting, we train five students with different seeds. Since there is some variance in students' performance (we hypothesize due to the small training sets) we report the **median** and **interquantile range (IQR)** around it (relative to the 25-75 percentile).

---

[3]While scalar mixing reduced variance of student performance, SMaT also worked with other common pooling methods.

Table 1: Results for the IMDB dataset with respect to student *simulability* in terms of accuracy (%). *Underlined* values indicate higher simulability than baseline with non-overlapping IQR.

|  | 500 | 1,000 | 2,000 |
|---|---|---|---|
| No Explainer | 81.72 [81.24:81.75] | 83.44 [83.36:83.63] | 84.84 [84.80:84.88] |
| Gradient L2 | 81.66 [81.32:82.00] | 82.98 [82.72:83.08] | 84.78 [84.96:85.08] |
| Gradient × Input | 84.83 [84.79:84.88] | 81.15 [80.95:81.36] | 83.84 [83.59:84.99] |
| Integrated Gradients | 82.99 [82.59:82.99] | 81.79 [81.72:81.87] | 84.20 [84.03:85.03] |
| Attention (*all layers*) | 83.00 [82.60:83.00] | 85.72 [85.72:86.23] | 90.08 [89.72:90.11] |
| Attention (*last layer*) | 80.91 [79.99:81.07] | 83.15 [82.91:83.51] | 91.47 [91.39:91.56] |
| Attention (**SMaT**) | **91.48** [91.40:91.56] | **92.56** [92.28:92.83] | **92.84** [92.84:93.08] |

## 5.1   Text Classification

For text classification, we consider the IMDB dataset [Maas et al., 2011], a binary sentiment classification task over highly polarized English movie reviews. As the base pretrained model, we use the small ELECTRA model [Clark et al., 2020], with 12 layers and 4 heads in each (total 48 heads). Like the setting in Pruthi et al. [2020], we use the original training set with 25,000 samples to train the teacher, and further split the test set into a training set for the student and a dev and test set. We vary the number of samples the student is trained on between 500, 1,000, and 2,000. We evaluate *simulability* using accuracy (i.e., what percentage of student predictions match with teacher predictions). The teacher model obtains 91% accuracy on the student test set.

Table 1 shows the results in terms of simulability (Equation 1) for the three settings. We can see that, overall, the attention explainer trained with SMaT leads to students that simulate the teacher model much more accurately than students trained without any explanations, and more accurately than students trained with any *static* explainer across all student training set sizes. Interestingly, the gradient-based explainers only improve over the baseline students when the amount of training data is very low, and actually degrade simulability for larger amounts of data (see discussion in A). Using only heads from the last layer seems to have the opposite effect, leading to higher simulability than all other static explainers only for larger training sets.

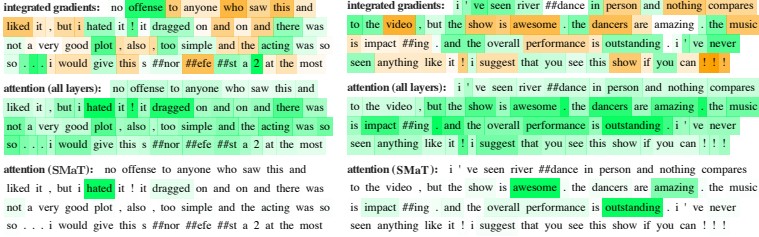

Figure 3: Explanations given by integrated gradients, attention (*last layer*), and our learned attention explainer (SMaT) for two movie reviews of the IMDB dataset (negative and positive examples). Green and orange represent positive and negative contributions, respectively.

Table 2: *Plausibility* on *MovieReviews* in terms of AUC. * represents methods that use human labels.

|  | AUC |
|---|---|
| Grad. L2 | 0.65 |
| Grad. × Input | 0.51 |
| Integrated Grad. | 0.53 |
| Attn. (*all layers*) | 0.68 |
| Attn. (*last layer*) | 0.61 |
| Attn. (**SMaT**) | **0.73** |
| Attn. (*best layer*)* | 0.75 |
| Attn. (*best head*)* | 0.75 |

**Plausibility analysis.** We select the median model trained with 1,000 samples and extract explanations for test samples from the MovieReviews dataset [DeYoung et al., 2020], which contains binary sentiment movie reviews from Rotten Tomatoes alongside human-rationale annotation. Since the labels are binary (indicating whether a token is part of the explanation or not) and the predicted scores are real values, we follow Fomicheva et al. [2021a] and report our results in terms of the Area Under the Curve (AUC), which automatically considers multiple binarization thresholds. The results are shown in Table 2 along with two randomly selected examples of extracted explanations in Figure 3. We found that gradient-based explanations are less plausible than those using attention (with the exception of *Grad. L2*, which is similar to static attention) and that ones produced with SMaT achieve the highest plausibility, indicating that our learned explainer can produce human-like explanations while maximizing simulability. Moreover, SMaT achieves a similar AUC score to the best performing attention layer and head,[4] while not requiring *any* human annotations. This is evidence that scaffolded simulability, while not explicitly designed for it, is a good proxy for plausibility and "human-like" explanations.

---

[4]AUC scores obtained by independently trying all attention heads and layers of the model.

Table 3: *Simulability* results, in terms of accuracy (%), on the CIFAR100 dataset. *Underlined* values represent better performance than baseline with non-overlapping IQR

|  | 2,250 | 4,500 | 9,000 |
|---|---|---|---|
| No Explainer | 81.16 [80.98:81.26] | 84.02 [83.98:84.24] | 85.20 [85.17:85.26] |
| Gradient L2 | 80.97 [80.91:81.10] | 83.98 [83.81:84.23] | 85.13 [84.97:85.50] |
| Gradient × Input | 80.93 [80.82:81.04] | 83.99 [83.98:84.13] | 85.33 [84.85:85.35] |
| Integrated gradients | 80.22 [80.17:80.35] | 83.44 [83.25:83.44] | 84.99 [84.76:85.22] |
| Attention (*all layers*) | 82.53 [82.53:82.62] | 84.81 [84.74:84.92] | **85.92** [85.78:85.94] |
| Attention (*last layer*) | 82.34 [82.30:82.60] | 84.65 [84.56:84.81] | 85.31 [84.84:85.31] |
| Attention (**SMaT**) | **83.09** [82.77:83.28] | **85.42** [85.39:85.85] | **85.96** [85.74:86.35] |

## 5.2 Image Classification

To validate our framework across multiple modalities, we consider image classification on the CIFAR-100 dataset [Krizhevsky, 2009]. We use as the base model the Vision Transformer (ViT) [Dosovitskiy et al., 2020], in particular the base version with $16 \times 16$ patches that was only pretrained on ImageNet-21k [Ridnik et al., 2021]. We up-sample images to to a $224 \times 224$ resolution.

Since the self-attention mechanism in the ViT model only works with patch representations, the explanations produced by attention-based explainers will be at patch-level rather than pixel-level. We split the original CIFAR-100 training set into a new training set with 45,000 and a validation set with 5,000. Unlike the previous task, we reuse the training set for both the teacher and student, varying the number of samples the student is trained with between 2,250 (5%), 4,500 (10%) and 9,000 (20%). We use accuracy as the simulability metric and the teacher obtains 89% on test set.

Table 3 shows the results for the three settings. Similarly to the results in the text modality, the attention explainer trained with SMaT achieves the best scaffolding performance, although the gaps to static attention-based explainers are smaller (especially when students are trained with more samples). Here, the gradient-based explainers always degrade simulability across the tested training set sizes and and it seems important that the explanations include attention information from layers other than the last one.

**Plausibility analysis.** Since there are no available human annotations for plausibility in the CIFAR-100 dataset, we design a user study to measure the plausability of the considered methods. The original image and explanations extracted with Gradient × Input, Integrated Gradients, Attention (*all layers*), and Attention (SMaT) are shown to the user, and the user has to rank the different explanations to answer the question *"Which explanation aligns the most with how you would explain a similar decision?"*. Explanations were annotated by three volunteers. After collecting results, we compute the *rank* and the *TrueSkill* rating [Herbrich et al., 2007] for each explainer (roughly, the "skill" level if the explainers where players in game). Further description can be found in Appendix B. The results are shown in Table 4. As in the previous task, attention trained with SMaT outperforms all other explainers in terms of plausibility, and its predicted *rating* is much higher than all other explainers. We also show examples of explanations for a set of randomly selected images in Figure 4.

Input image Integ. Grad. Attn. (all lx.) Attn. (SMaT) Input image Integ. Grad. Attn. (all lx.) Attn. (SMaT)

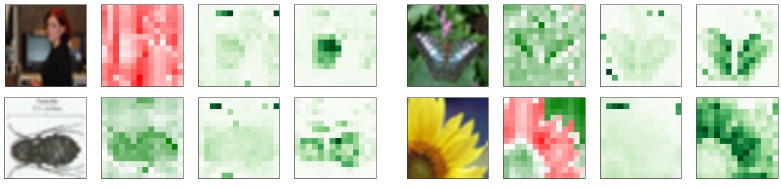

Figure 4: Explanations given by integrated gradients, attention (*all layers*), and learned attention explainer for a set of input images of CIFAR-100. Gold labels are: "television", "butterfly", "cockroach", and "sunflower".

Table 4: *Plausibility* results of the human study on visual explanations.

|  | Rank | *TrueSkill* |
|---|---|---|
| Grad. × Input | 3-4 | -2.7±.67 |
| Integ. Grad. | 3-4 | -2.1±.67 |
| Attn. (*all lx.*) | 2 | 0.7±.67 |
| Attn. (**SMaT**) | **1** | **4.3±.70** |

## 5.3 Machine Translation Quality Estimation

Quality Estimation (QE) is the task of predicting a quality score given a sentence in a source language and a translation in a target language from a machine translation system, which requires models that consider interactions between the two inputs, source and target. Scores tend to be continuous values (making this a regression task) that were collected from expert annotators.

Table 5: *Simulability* results, in terms of Pearson correlation, on the ML-QE dataset. *Underlined* values represent better performance than baseline with non-overlapping IQR.

|  | 2,100 | 4,200 | 8,400 |
|---|---|---|---|
| No Explainer | .7457 [.7366:.7528] | .7719 [.7660:.7802] | .7891 [.7860:.7964] |
| Gradient L2 | .8065 [.8038:.8268] | .8535 [.7117:.8544] | **.8638** [.8411:.8657] |
| Gradient × Input | .6846 [.6781:.6894] | .6922 [.6885:.6965] | .7141 [.7136:.7147] |
| Integrated gradients | .6686 [.6677:.6694] | .7086 [.6994:.7101] | .7036 [.6976:.7037] |
| Attention (*all layers*) | .8120 [.7955:.8125] | .8193 [.8186:.8280] | .8467 [.8464:.8521] |
| Attention (*last layer*) | .7486 [.7484:.7534] | .7720 [.7672:.7726] | .7798 [.7717:.7814] |
| Attention (**SMaT**) | **.8156** [.8096:.8183] | **.8630** [.8412:.8724] | **.8561** [.8512:.8689] |

Table 6: Plausibility results for source and target inputs for each language pair of the MLQE-PE dataset in terms of AUC. * represents *supervised* methods that use human labels in some form.

|  | EN-DE | | EN-ZH | | ET-EN | | NE-EN | | RO-EN | | RU-EN | | OVERALL | |
|---|---|---|---|---|---|---|---|---|---|---|---|---|---|---|
|  | src. | tgt. | src. | tgt. | src. | tgt. | src. | tgt. | src. | tgt. | src. | tgt. | src. | tgt. |
| Gradient L2 | **0.64** | **0.65** | 0.65 | 0.49 | **0.67** | 0.61 | **0.68** | **0.55** | **0.72** | 0.68 | **0.65** | 0.54 | **0.67** | 0.59 |
| Gradient × Input | 0.58 | 0.60 | 0.61 | 0.51 | 0.60 | 0.54 | 0.61 | 0.49 | 0.64 | 0.59 | 0.58 | 0.51 | 0.61 | 0.54 |
| Integrated Gradients | 0.59 | 0.60 | 0.63 | 0.49 | 0.60 | 0.52 | 0.64 | 0.48 | 0.64 | 0.59 | 0.60 | 0.51 | 0.62 | 0.53 |
| Attention (*all layers*) | 0.60 | 0.63 | **0.68** | **0.52** | 0.60 | 0.61 | 0.58 | **0.55** | 0.66 | **0.70** | 0.62 | **0.55** | 0.62 | 0.59 |
| Attention (*last layer*) | 0.51 | 0.49 | 0.61 | 0.49 | 0.51 | 0.50 | 0.55 | 0.48 | 0.52 | 0.57 | 0.56 | 0.50 | 0.54 | 0.50 |
| Attention (**SMaT**) | **0.64** | **0.65** | **0.68** | **0.52** | 0.66 | **0.64** | 0.66 | 0.54 | 0.71 | **0.70** | 0.61 | 0.54 | 0.66 | **0.60** |
| Attention (*best layer*)* | 0.64 | 0.65 | 0.69 | 0.64 | 0.64 | 0.68 | 0.68 | 0.68 | 0.71 | 0.76 | 0.64 | 0.59 | 0.65 | 0.65 |
| Attention (*best head*)* | 0.67 | 0.67 | 0.70 | 0.65 | 0.70 | 0.70 | 0.70 | 0.69 | 0.73 | 0.75 | 0.67 | 0.60 | 0.67 | 0.66 |

Interpreting quality scores of machine translated outputs is a problem that has received recent interest [Fomicheva et al., 2021a] since it allows identifying which words were responsible for a bad translation. We use the MLQE-PE dataset [Fomicheva et al., 2020], which contains 7,000 training samples for each of seven language pairs alongside word-level human annotation. We use as the base model a pretrained XLM-R-base [Conneau et al., 2019], a multilingual model with 12 layers and 12 heads in each (total of 144 heads).

We exclude one of the language pairs in the dataset (si-en) since the XLM-R model did not support it, leading to a training set with 42,000 samples. Similar to the CIFAR-100 case, we reuse the same training set for both the teacher and student, sampling a subset for the latter. We vary the number of samples the student is trained with between 2,100 (5%), 4,200 (10%) and 8,400 (20%). Since this is a regression task, we evaluate simulability using the Pearson correlation coefficient between student and teacher's predictions.[5] The teacher achieves 0.63 correlation on the test set.

Table 5 shows the results for the three settings. Similar to other tasks, the attention explainer trained with SMaT leads to students with higher simulability than baseline students and similar or higher than *static* explainer across all training set sizes. Curiously, the *Grad. L2* explainer achieves very high simulability for this task. It even has a higher *median* simulability score than SMaT for 8,400 samples. However, we attribute this to variance in the student training set sampling (that could lead to an imbalance in language pair proportions) which could explain why SMaT performance degrades with more samples. For this task, the gradient-based explainers always degrade simulability across the tested training set size. It also seems that using only the last layer's attention is also ineffective at teaching students, achieving the same performance as the baseline.

**Plausibility analysis.** We select the median model trained with 4,200 samples and follow the approach devised in the Explainable QE shared task to evaluate plausibility [Fomicheva et al., 2021a], which consists of evaluating the human-likeness of explanations in terms of AUC only on the subset of translations that contain errors. The results are shown in Table 6. We note that for all language pairs, SMaT performs on par or better than static explainers, and only being surpassed by *Grad. L2* in the *source-side* over all languages. Comparing with the best attention layer/head, an approach used by Fomicheva et al. [2021b], Treviso et al. [2021], SMaT achieves similar AUC scores for source explanations, but lags behind the best attention layer/head for target explanations on *-EN language pairs. However, as stressed previously for text and image classification, SMaT sidesteps human annotation and avoids the cumbersome approach of independently computing plausibility scores for all heads.

---

[5]Pearson correlation is the standard metric used to evaluate sentence-level QE models.

# 6 Importance of the Head Projection

A major component of our framework is the normalization of the head coefficients, as defined in § 4. Although many functions can be used to map scores to probabilities, we found empirically that SPARSEMAX performs the best, while other transformations such as SOFTMAX and 1.5-ENTMAX [Peters et al., 2019], a sparse transformation more dense than sparsemax, usually lead to poorly performing students (see Table 7).

Table 7: *Simulability* results, in terms of accuracy (%), on the MLQE dataset with 4200 training examples, with different normalization functions.

|  | SPARSEMAX | SOFTMAX | 1.5-ENTMAX | No Normalization |
|---|---|---|---|---|
| No Explainer | .7719 ± [.7660:.7802] | .7719 ± [.7660:.7802] | .7719 ± [.7660:.7802] | .7719 ± [.7660:.7802] |
| Attention (*all layers*) | .8193 ± [.8186:.8280] | .7345± [.7335:.7390] | .7152 ± [.7111:.7161] | .7781 ± [.7762:.7791] |
| Attention (*last layer*) | .7720 ± [.7672:.7726] | .7697± [.7659:.7715] | .7807 ± [.7652:.7821] | .7768 ± [.7764:.7807] |
| Attention (**SMaT**) | **.8630** ± [.8412:.8724] | .7439 ± [.7430:.7484] | .7163 ± [.7130:.7239] | .8002 ± [.7919:.8100] |

Furthermore, another benefit of SPARSEMAX is that it produces a small subset of *active* heads. The heatmaps of attention coefficients ($\lambda_T$) learned after training, shown in Figure 5, exemplify this. We can see that the dependency between head position (layer it belongs to) and its coefficient is task/dataset/model specific, with CIFAR-100 and MLQE having opposite observations. We also found empirically that *active* heads ($\lambda_T^h > 0$) usually lead to higher plausibility scores than *inactive* heads, further reinforcing the good plausibility findings of SMaT.

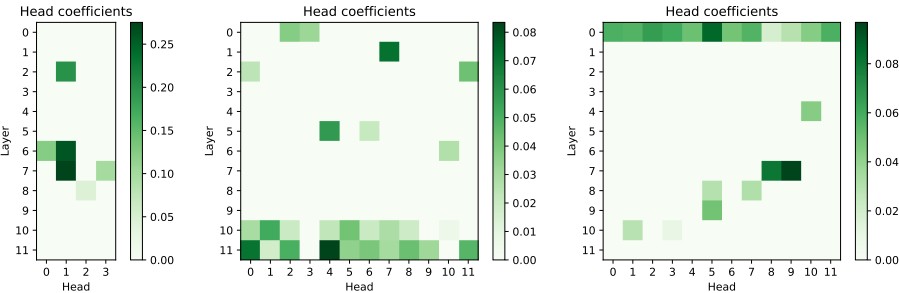

Figure 5: Head coefficients for text classification (left), image classification (middle), and quality estimation (right), illustrating that only a small subset of attention heads are deemed relevant by SMaT due to SPARSEMAX.

# 7 Related Work

**Explainability for text & vision.** Several works propose explainability methods to interpret decisions made by NLP and CV models. Besides gradient and attention-based approaches already mentioned, some extract explanations by running the models with perturbed inputs [Ribeiro et al., 2016, Feng et al., 2018, Kim et al., 2020]. Others even define custom backward passes to assign relevance for each feature [Bach et al., 2015]. These methods are commonly employed together with post-processing heuristics, such as selecting only the top-k tokens/pixels with higher scores for visualization. Another line of work seeks to build a classifier with inherently interpretable components, such as methods based on attention mechanisms and rationalizers [Lei et al., 2016, Bastings et al., 2019].

**Evaluation of explainability methods.** As mentioned in the introduction, early works evaluated explanations based on properties such as *consistency*, *sufficiency* and *comprehensiveness*. Jacovi and Goldberg [2020] recommended the use of a graded notion of faithfulness, which the ERASER benchmark quantifies using the idea of sufficient and comprehensive rationales, alongside compiling datasets with human-annotated rationales for calculating plausibility metrics [DeYoung et al., 2020]. Given the disagreement between explainability methods, Neely et al. [2021] showed that without a faithful ground-truth explanation it is impossible to determine which method is better. Diagnostic tests such as the ones proposed by Adebayo et al. [2018], Wiegreffe and Pinter [2019] and Atanasova et al. [2020] are more informative yet they do not capture the main goal of an explanation: the ability to communicate an explanation to a practitioner.

**Simulability.** A new dimension for evaluating explainability methods relies on the forward prediction/simulation proposed by Lipton [2016] and Doshi-Velez and Kim [2017], which states that humans should be able to correctly simulate the model's output given the input and the explanation. Chandrasekaran et al. [2018], Hase and Bansal [2020], Arora et al. [2022] analyze simulability via human studies across text classification datasets. Treviso and Martins [2020] designed an automatic framework where students (machine or human) have to predict the model's output given an explanation as input. Similarly, Pruthi et al. [2020] proposed the simulability framework that was extended in our work, where explanations are used to regularize the student rather than passed as input.

**Learning to explain.** The concept of simulability also opens a path to learning explainers. In particular Treviso and Martins [2020] learn an attention-based explainer that maximizes simulability. However, directly optimizing for simulability sometimes led to explainers that learned trivial protocols (such as selecting only punctuation symbols or stopwords to leak the label). Our approach of optimizing a teacher-student framework is similar to approaches that optimize for model distillation [Zhou et al., 2021]. However, these approaches modify the original model rather than introduce a new explainer module. Raghu et al. [2021] propose a framework similar to ours for learning *commentaries* for inputs that speed up and improve the training of a model. However commentaries are model-independent and are optimised to improve performance on the real task. Rationalizers [Chen et al., 2018, Jacovi and Goldberg, 2021, Guerreiro and Martins, 2021] also directly learn to extract explanations, but can also suffer from trivial protocols.

# 8    Conclusion & Future Work

We proposed **SMaT**, a framework for directly optimizing explanations of the model's predictions to improve the training of a student *simulating* the said model. We found that, across tasks and domains, explanations learned with SMaT both lead to students that simulate the original model more accurately and are more aligned with how people explain similar decisions when compared to previously proposed methods. On top of that, our parameterized attention explainer provides a principled way for discovering relevant attention heads in transformers.

Our work shows that scaffolding is a suitable criterion for both evaluating and optimizing explainability methods, and we hope that SMaT paves way for new research to develop expressive interpretable components for neural networks that can be directly trained without human-labeled explanations. However, it should be noted that "interpretability" is a loosely defined concept, and therefore caution should be exercised when making statements about the quality of explanations based only on simulability, especially if these statements might have societal impacts.

We only explored learning attention-based explainers, but our method can also be used to optimize other types of explainability methods, including gradient-based ones, by introducing learnable parameters in their formulations. Another promising future research direction is to explore using SMaT to learn explanations other than saliency maps.

## Acknowledgments

This work was supported by the European Research Council (ERC StG DeepSPIN 758969), by EU's Horizon Europe Research and Innovation Actions (UTTER, contract 101070631), by P2020 project MAIA (LISBOA-01-0247- FEDER045909), and Fundação para a Ciência e Tecnologia through project PTDC/CCI-INF/4703/2021 (PRELUNA) and contract UIDB/50008/2020. We are grateful to Nuno Sabino, Thales Bertaglia, Henrico Brum, and Antonio Farinhas for the participation in human evaluation experiments.

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
