## A    Explainer Details

With the *integrated gradients* explainer [Sundararajan et al., 2017], we use 10 iterations for the integral in the *simulability* experiments (due to the computation costs) and 50 iterations for the *plausability* experiments. We use zero vectors as baseline embeddings, since we found little variation in changing this. For both gradients-based explainers, we project into the simplex by using the SOFTMAX function, similar to the attention-based explainers. This results in very negative values having low probability values. Moreover, for evaluating plausibility on text classification and translation quality estimation, we computed the explanation score of a single word by summing the scores of its word pieces.

We would like to note that, unlike the setting in Pruthi et al. [2020], we **do not** apply a *top-k* post-processing heuristic on gradients/attention logits, instead directly projecting them to the simplex. This might explain the difference in results to the original paper, particularly for the low simulability performance of static explainers.

# B Human Study for Visual Explanations

The annotations were collected through an annotation webpage, built on top of Flask. Figure 6 shows the three pages of the site. During the annotation, users were asked to rank four explanations, unnamed and in random order. After collecting the ratings, we computed the *TrueSkill* rating, with an initial rating for each method of $\mu = 0, \sigma = 0.5$. After learning the ratings, we then compute the *ranks* by obtaining the 95% confidence interval for the rating each method, and constructing a partial ordering of methods based on this.

The volunteers were a mixture of graduates or graduate students known by the authors. However we would like to point out that due to blind nature of the method annotation, the chance of bias is low.

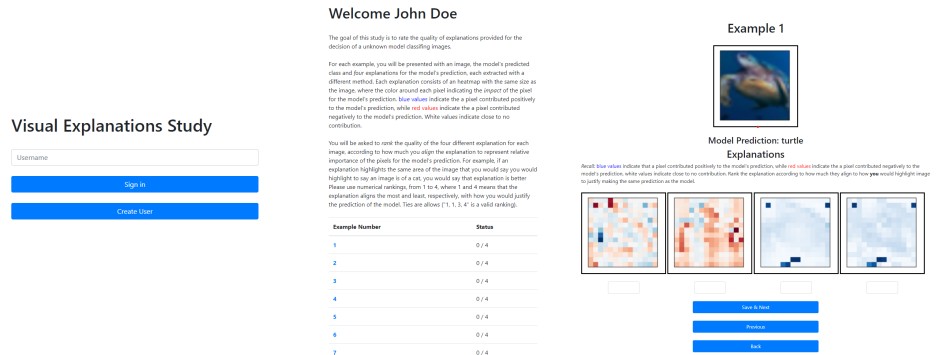

Figure 6: Login page (left), dashboard (middle) and annotation page (right)

In Figures 7 and 8 we show raw attention explanations extracted from all layers (rows) and heads (columns) of the teacher transformer used in our CIFAR-100 experiments. Cross-checking the explanations with the most relevant heads selected by SMaT for image classification, we can see that most selected heads produce plausible explanations (e.g., attention heads from the last layers).

Original image ("television")

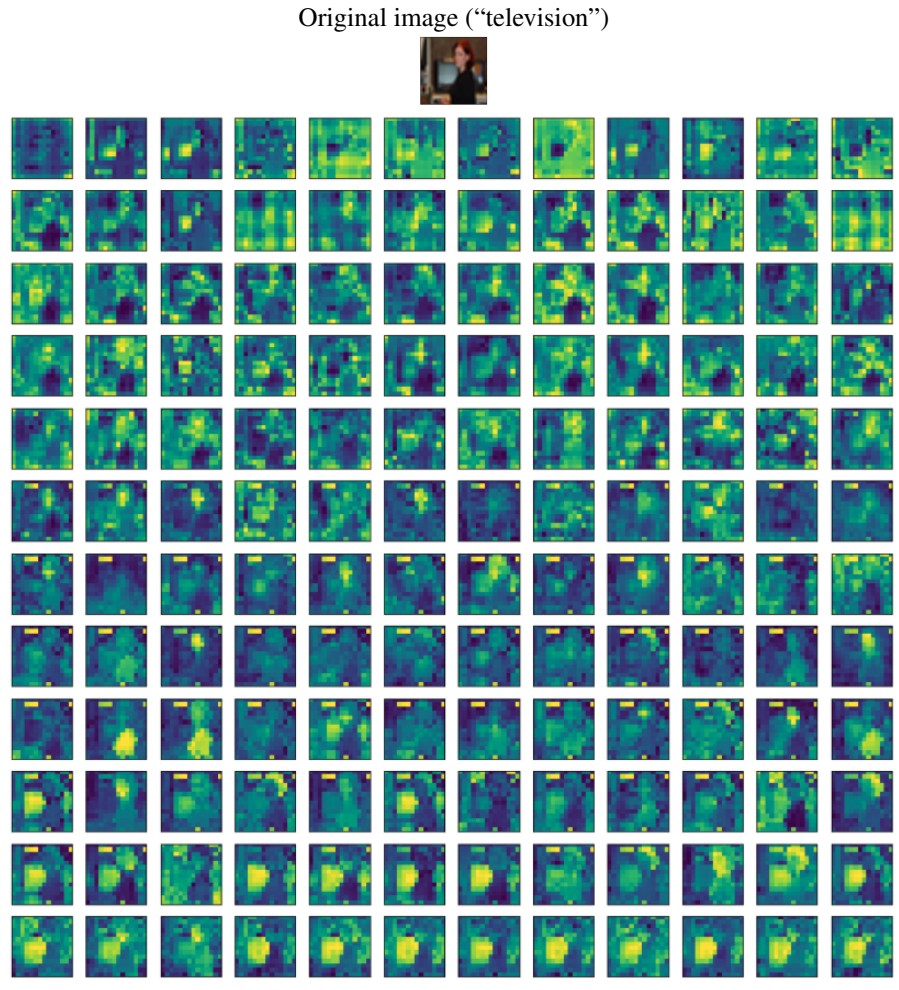

Figure 7: Explanations from all layers (rows) and heads (columns) of the CIFAR-100 teacher model.

Original image ("butterfly")

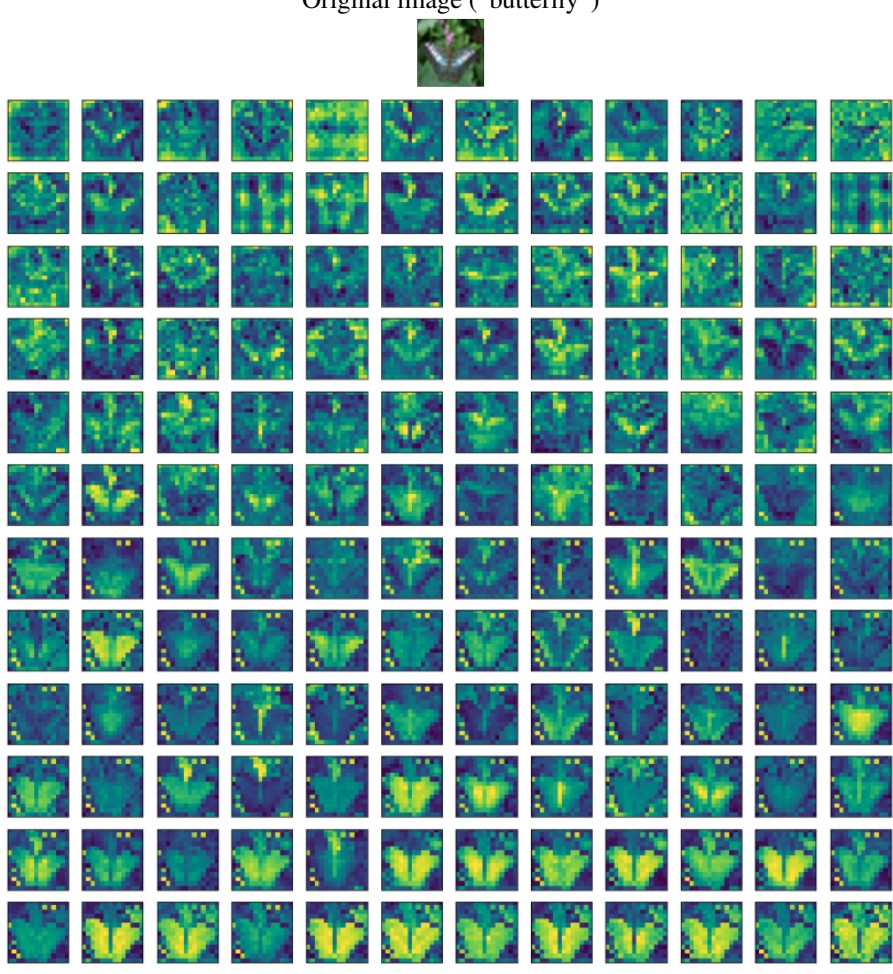

Figure 8: Explanations from all layers (rows) and heads (columns) of the CIFAR-100 teacher model.