# OpenReview forum: "Learning to Scaffold: Optimizing Model Explanations for Teaching"
_NeurIPS.cc/2022/Conference — NeurIPS 2022 Accept_

### Official Review · Reviewer_b9Jw · 2022-07-12

**Rating:** 8
**Confidence:** 2
**Soundness:** 4 excellent
**Presentation:** 4 excellent
**Contribution:** 3 good

**Summary:**

The authors introduce SMaT (Scaffold-Maximizing Training),  a framework to directly optimize the explanations produced on a teacher model to improve the distillation of a student model simulating the teacher model (and creating an explainer for the student model too). The authors limit the architecture of the models to attention based transformers, and conduct experiments on three tasks (nlp, translation, image classification). Using SMaT resulted in students which more accurately simulate the teacher and produces more plausible explanations.

**Questions:**

- Would SMaT work in the case of a born-again network? In all the examples the student is limited to a smaller set of data.

**Limitations:**

The authors do not address this matter.

**Strengths And Weaknesses:**

The paper presents an interesting approach to learning explainers in such a way as to more accurately distill a teacher to a student model. The authors demonstrate their results over three very different tasks. The authors point show when other explainability approaches (Gradient L2 for example), results in improved simulability (often when more data is made available). The paper is well written and the experiment is well designed.

---

> ### Author Response · Authors · 2022-07-31
> **Response**
>
> > Would SMaT work in the case of a born-again network? In all the examples the student is limited to a smaller set of data.
>
> This is an excellent question. In the context of this work, our goal was to attain good explanations by optimizing both the explainer and the student without necessarily making the student model surpass the teacher. Our experiments hinted that the best explanations are obtained when the student is severely constrained in terms of data, since explanations help the most in this case. However, one future direction we are exploring is born-again-style multi-generational learning to understand if we can iteratively train explainers and students to be able to match or even surpass the performance of the teacher with less data. This also connects with R2’s comments on the connection with knowledge distillation.

---

> > ### Comment · Reviewer_b9Jw · 2022-08-10
> > **Multi-generational learning.**
> >
> > Thank you for addressing my question. I look forward to your results in these further experiments.

---

### Official Review · Reviewer_Z8bs · 2022-07-13

**Rating:** 6
**Confidence:** 3
**Soundness:** 3 good
**Presentation:** 3 good
**Contribution:** 3 good

**Summary:**

The paper proposes a bi-level optimization procedure to directly optimize the simulability objective of explanability. In order for the optimization to work well, the paper proposes a new parametrized attention layer that is a weighted sum of the attention heads from Transformer models. This new layer (SMaT) works very well under this objective, showing high simulability (expected) and plausibility rating over other saliency-based methods.

**Questions:**

- Can you comment on what is the student model? Is it only constrained in a way that it has less parameters than the teacher, but otherwise receive the same input and have the same architecture as the teacher? Can your work be viewed as a type of a model-compression/knowledge distillation?
- There is a type of work that optimizes through reinforce to hard-select words that shows to have high precision at selecting words that are also selected from human rationale [3]. This also referred to as "hard attention" -- have you guys considered showing a baseline of this method for AUC on MovieReviews? Is it better than SMaT or worse?

**Strengths And Weaknesses:**

Overall comment:

I'm generally impressed by this paper. Explainability work has seen a promising set of development in the last 3 years focusing specifically on the "why" and "so what" question of explainability -- why do we want explainability and what should explainability do (functionally). I'm excited to see that this paper takes a step in that direction focusing on pushing the boundary on these new properties of explanations. However, though generally well-written, I'm a bit concerned with a few choices the authors took. First of all, the explainers being differentiable. This significantly limits the type of explainers we can use, especially the game-theoretic ones. Attention is not explanation [1] because it does not faithfully represent the output of the model (and does not conform to the axioms of explainability [2]). The authors did use Integrated Gradients -- which mitigates this part of my criticism. Second concern, since the learned attention is the "winning" explainer model, I wonder if this is because **attention is the superior explanation method** or because **attention explainer is easier to optimize under the bi-level optimization objective the paper proposed** (aka., any method that can maximize this objective will be a good explanation method, regardless on whether they satisfy other criteria of explainability such as completeness). This is probably a very hard question to address or to design experiments to falsify,  but would like to hear authors' thoughts on this.

Strengths:
- The paper is clear and well-written
- The experiment is very thorough (including all three major ML application fields)
- Plausibility is investigated through either human annotations or user studies
- The intention to directly optimize

Weaknesses:
- Some philosophical questions asked above.
- Could compare to some other interpretability baselines like shapley values (but this is not my main concern, and nor should we require every single paper to run experiment on every method that the reviewer happens to know)

[1] Jain, S., & Wallace, B. C. (2019). Attention is not explanation. arXiv preprint arXiv:1902.10186.
[2] Sundararajan, M., Taly, A., & Yan, Q. (2017, July). Axiomatic attribution for deep networks. In International conference on machine learning (pp. 3319-3328). PMLR.
[3] Lei, T., Barzilay, R., & Jaakkola, T. (2016). Rationalizing neural predictions. arXiv preprint arXiv:1606.04155.

---

> ### Author Response · Authors · 2022-07-31
> **Response**
>
> >  I'm a bit concerned with a few choices the authors took. First of all, the explainers being differentiable. This significantly limits the type of explainers we can use, especially the game-theoretic ones.
>
> Our explainers should indeed be differentiable in order for our framework to work. However, if we consider them as just black-box, un-parametrized explainers, our framework can still be used with explanations that combine these different methods, and the bi-level optimization procedure can be used to learn the weight of each of these explainers. This is a direction we are excited about exploring in future work. However, we acknowledge that for non-differentiable explainers, this would require a cumbersome computation and also that it is not trivial to find a good parameterization.
>
>
> > Since the learned attention is the "winning" explainer model, I wonder if this is because attention is the superior explanation method or because attention explainer is easier to optimize under the bi-level optimization objective the paper proposed.
>
> We strongly believe that attention **is not** the superior explainability method, as several factors impact their benefits and drawbacks, and also that our bi-level optimization **does not** favor attention explainers. For example, in Table 5 we can see that Gradient L2 achieved very high results, surpassing attention for $N \geq 4200$, while still obtaining high plausibility scores for most language pairs in Table 6. Nevertheless, we do believe attention is a strong method since it not only achieves high plausibility scores in all of our experiments, but it can also be easily parameterized and pushed further to improve its simulability.
>
>
> > Can you comment on what is the student model? Is it only constrained in a way that it has less parameters than the teacher, but otherwise receive the same input and have the same architecture as the teacher?
>
> The student is a model that tries to match the original predictions of the teacher in a constrained setting. Note that a constrained student does not necessarily have less capacity but could also just be trained on a much smaller subset of training data. The latter is precisely what we do in this work. For example, in the IMDb experiments we used the same pretrained model for the student as we used for the teacher (ELECTRA), but with 5-20% of the data.
>
> > Can your work be viewed as a type of a model-compression/knowledge distillation?
>
> Our setup reduces to hard-label knowledge distillation when we remove the explainer regularizer $L\_{expl}$ in the simulability loss (Equation 3) and to soft-label knowledge distillation when it is replaced by a KL-loss between output distributions. A significant difference is that we use the student just as an instrument, as our goal is to obtain better explainers. At the same time, accurate students emerge as by-products of this process. This connection was mentioned in the related works section. We will try to clarify this connection further in the final version.
>
>
> > Have you guys considered showing a baseline of this method ("hard attention") for AUC on MovieReviews? Is it better than SMaT or worse?
>
> We did not experiment with stochastic hard attention methods as they usually tend to be highly unstable to train (Guerreiro and Martins, 2021). However, we did try learning explainers with deterministic sparse attention, such as sparsemax and entmax. From our preliminary experiments, the results were not significantly different from using dense softmax attention, so we decided to stick with softmax attention explainers.
>
> ---
>
> Pruthi, Danish, Rachit Bansal, Bhuwan Dhingra, Livio Baldini Soares, Michael Collins, Zachary C. Lipton, Graham Neubig, and William W. Cohen. "Evaluating Explanations: How much do explanations from the teacher aid students?." TACL (2022).
>
> Guerreiro, Nuno Miguel, and André FT Martins. "Spectra: Sparse structured text rationalization." EMNLP (2021).

---

### Official Review · Reviewer_HtBu · 2022-07-17

**Rating:** 8
**Confidence:** 4
**Soundness:** 3 good
**Presentation:** 3 good
**Contribution:** 2 fair

**Summary:**

This work introduces Scaffold-Maximizing Training, a process for learning better “explanations” for learned models by optimizing the effect the produced explanations have on downstream “simulatability.” There’s a lot of terminology and nuance to the arguments in this paper, but in reading through it, my understanding is as follows:

- Given a trained model T (off the shelf, black-box, e.g., an image/text classifier), we want to be able to generate good explanations of why it predicted a certain output.
- How do we know what a “good explanation” is? The field hasn’t yet converged to a single metric, but one plausible measure that seems to be well-adopted and well-motivated is the *simulability* of the trained model T (or teacher model); in other words, how well can we use the explanations to train a student model (lower capacity than the teacher T) that “simulates” how the teacher model T performs on held-out data.
- The idea is that a student that perfectly simulates the teacher T both in terms of prediction, *and in terms of explanations w/ its own explanation model* is a sign that the original explanations for T are good!

In general, the paper builds up this narrative cleanly, ultimately arriving at a bi-level optimization problem for learning the student model parameters (as well as the student model’s explanation model), followed by the teacher model’s explanation model as the outer loop. Experiments on image classification, text classification, and multilingual regression (translation quality estimation) show that explainers trained via SMaT lead to higher simulability scores and “plausibility” (human likeness of explanations) than methods that just statically condition on some external explanation process, or that don’t use explanations at all.

**Questions:**

- Following from the last paragraph of weaknesses — what happens if you remove the explanation regularization from the objective for all of the static and learned models (basically ablate the student explainer model altogether). At this point, I don’t know if the better simulability is actually a result of the bi-level optimization procedure (learning better explanations) or just learning to regress explanations with a student model...


**Limitations:**

This paper is fairly clear about its limitations, acknowledging that the “explanation” methods used in this work are mostly saliency driven (attention maps). Like the paper suggests, I’d be interested in seeing applications of SMaT to other explainability approaches!


**Strengths And Weaknesses:**

The strengths of this paper are in it’s clarity and originality; I strongly believe that each part of the paper is well-motivated, and seeks to build to an answer of the question — what makes better explanations? The experiments are comprehensive and the results while not overwhelmingly convincing, do make enough of a point that the SMaT protocol is effective.

However, the biggest weakness of the paper is the assumption driving the main optimization objective of the framework — the inner-level optimization not only optimizes the simulability score of the learned student model relative to the teacher, but the “explanation regularization loss” between the explanations produced under the student and the explanations produced under the teacher. It seems to me that this is an incredibly strong assumption, that doesn’t immediately fall out of the logic from the paper — if the goal is to produce better explanations via simulability, and optimize the explanations for better simulability, then this sort of “explanation matching” loss is not necessary?

EDIT: Rebuttal cleared this up!

---

> ### Author Response · Authors · 2022-07-31
> **Response**
>
> > (...) the inner-level optimization not only optimizes the simulability score of the learned student model relative to the teacher, but the “explanation regularization loss” between the explanations produced under the student and the explanations produced under the teacher. It seems to me that this is an incredibly strong assumption, that doesn’t immediately fall out of the logic from the paper.
>
> We would like to clarify a potential misunderstanding in the role of explanations in the inner and outer optimization loops. Simulability, as defined by Pruthi et al. (2022), is measured by the performance of a student model on a held-out dataset **without** explanations (or explanation matching as in our inner loop setup). This means that explanations are used to help train the student but not to evaluate it. This can be seen, for example, in Equations 1 and 5 since no explainer appears in the simulability and its loss, therefore the only assumption we make is that explanations aid the student learning process. To summarize, regularizing the objective through the form of explanation matching provides a way of integrating information from the explanations during the student training without making the model reliant on explanations.
>
> > If the goal is to produce better explanations via simulability, and optimize the explanations for better simulability, then this sort of “explanation matching” loss is not necessary?”
> (...)
> Following from the last paragraph of weaknesses — what happens if you remove the explanation regularization from the objective for all of the static and learned models (basically ablate the student explainer model altogether). At this point, I don’t know if the better simulability is actually a result of the bi-level optimization procedure
>
> If we remove explanations from the objective (e.g., by removing the explanation regularization term in Equation 3), then $\mathcal{L}\_{student}$ reduces to
> $\mathcal{L}\_{sim}$ and consequently $\phi\_S$ and $\phi\_T$ can be dropped from Equations 4 and 5,  arriving at a hard-label knowledge distillation, which is our baseline. As our and Pruthi et al. (2022) results show, the explainer regularization term allows the student to learn something about the teacher's decision process and match its predictions on the test set (i.e., improve simulation accuracy). Therefore, to answer the question, it is necessary to have the explanation matching loss to produce better students.
>
> >This paper is fairly clear about its limitations, acknowledging that the “explanation” methods used in this work are mostly saliency driven (attention maps). Like the paper suggests, I’d be interested in seeing applications of SMaT to other explainability approaches!
>
> Yes, we are excited to extend our work to include different explainability approaches in the future. Saliency maps constitute a large fraction of all explainability approaches, and therefore we focus on those and provide evidence that our bi-level procedure is effective at producing good explanations across tasks and domains.
>
> ---
>
> Pruthi, Danish, Rachit Bansal, Bhuwan Dhingra, Livio Baldini Soares, Michael Collins, Zachary C. Lipton, Graham Neubig, and William W. Cohen. "Evaluating Explanations: How much do explanations from the teacher aid students?." TACL (2022).

---

> > ### Comment · Reviewer_HtBu · 2022-08-08
> > **Updated Score**
> >
> > Sorry for the delay in getting back to you! After your excellent rebuttal came out, I wanted to take a closer look at the prior work, and the assumptions made throughout those works (as well as revisiting this work in light of the clarifications).
> >
> > The framework here makes a lot more sense to me now, and I especially found that the way that you grounded the misconception I had in the existing baselines to be of great value. I have since updated my score, and believe this to be a strong paper.

---

### Author Response · Authors · 2022-07-31
**General Response**

We thank the reviewers for their positive comments and detailed feedback, and we are glad to learn that they appreciated the clarity and originality of our work, and found our approach to be well-motivated and effective.

Below, we respond to the raised questions.

---

### Meta-Review · Area_Chair_AtxZ · 2022-08-27

**Recommendation:** Accept
**Confidence:** Certain

**Metareview:**

This paper proposed a novel “Scaffold-Maximizing Training” framework to optimize model explanations, by leveraging a student model simulating the teacher model using the teacher’s explanations. The idea of constructing explanations such that student models can better simulate the teacher is interesting and appears to be an original contribution.

All reviewers agree that this paper made a solid contribution to the explainability of ML models, the experiments are comprehensive and thorough, and the paper is well written. Overall the paper is well-rounded with clear justifications of what each algorithmic component does, and convincing empirical support (the proposed algorithm has been demonstrated to be effective on a variety of tasks). After viewing the authors’ feedback, there is a broad consensus on accepting the paper.


**Award:**

No

---

### Decision · Program_Chairs · 2022-09-14

Accept